# Immunogenicity and reactogenicity of SARS-CoV-2 vaccines in people living with HIV in the Netherlands: A nationwide prospective cohort study

Kathryn S. Hensley[1], Marlou J. Jongkees[1☯], Daryl Geers[2☯], Corine H. GeurtsvanKessel[2], Yvonne M. Mueller[3], Virgil A. S. H. Dalm[3,4], Grigorios Papageorgiou[5], Hanka Steggink[6], Alicja Gorska[1], Susanne Bogers[2], Jan G. den Hollander[7], Wouter F. W. Bierman[8], Luc B. S. Gelinck[9], Emile F. Schippers[10,11], Heidi S. M. Ammerlaan[12], Marc van der Valk[13,14], Marit G. A. van Vonderen[15], Corine E. Delsing[16], Elisabeth H. Gisolf[17], Anke H. W. Bruns[18], Fanny N. Lauw[19], Marvin A. H. Berrevoets[20], Kim C. E. Sigaloff[21], Robert Soetekouw[22], Judith Branger[23], Quirijn de Mast[24], Adriana J. J. Lammers[25], Selwyn H. Lowe[26], Rory D. de Vries[2], Peter D. Katsikis[3], Bart J. A. Rijnders[1], Kees Brinkman[6], Anna H. E. Roukens[11‡], Casper Rokx[1‡]*

**1** Department of Internal Medicine, Section Infectious Diseases, and Department of Medical Microbiology and Infectious Diseases, Erasmus University Medical Centre, Rotterdam, Netherlands, **2** Department of Viroscience, Erasmus University Medical Centre, Rotterdam, Netherlands, **3** Department of Immunology, Erasmus University Medical Centre, Rotterdam, Netherlands, **4** Department of Internal Medicine, Division of Allergy & Clinical Immunology, Erasmus University Medical Centre, Rotterdam, Netherlands, **5** Department of Biostatistics, Erasmus University Medical Centre, Rotterdam, Netherlands, **6** Department of Internal Medicine and Infectious Diseases, OLVG Hospital, Amsterdam, Netherlands, **7** Department of Internal Medicine, Maasstad Hospital, Rotterdam, Netherlands, **8** Department of Internal Medicine, Section Infectious Diseases, University of Groningen, Groningen, Netherlands, **9** Department of Internal Medicine and Infectious Diseases, Haaglanden Medical Centre, The Hague, Netherlands, **10** Department of Internal Medicine, Haga Teaching Hospital, The Hague, Netherlands, **11** Department of Infectious Diseases, Leiden University Medical Centre, Leiden Netherlands, **12** Department of Internal Medicine, Catharina Hospital, Eindhoven, Netherlands, **13** Department of Internal Medicine and Infectious Diseases, DC Klinieken, Amsterdam, Netherlands, **14** Department of Infectious Diseases, Amsterdam Institute for Infection and Immunity, Amsterdam University Medical Centre, University of Amsterdam, Amsterdam, Netherlands, **15** Department of Internal Medicine, Medical Centre Leeuwarden, Leeuwarden, Netherlands, **16** Department of Internal Medicine and Infectious Diseases, Medisch Spectrum Twente, Enschede, Netherlands, **17** Department of Internal Medicine and Infectious Diseases, Rijnstate Hospital, Arnhem, Netherlands, **18** Department of Internal Medicine and Infectious Diseases, University Medical Centre Utrecht, Utrecht, Netherlands, **19** Department of Internal Medicine and Infectious Diseases, Medical Centre Jan van Goyen, Amsterdam, Netherlands, **20** Department of Internal Medicine, Elisabeth-Tweesteden Hospital, Tilburg, Netherlands, **21** Department of Internal Medicine, Division of Infectious Diseases, Amsterdam Institute for Infection and Immunity, Amsterdam University Medical Centre, Vrije Universiteit Amsterdam, Amsterdam, Netherlands, **22** Department of Internal Medicine and Infectious Diseases, Spaarne Gasthuis, Haarlem, Netherlands, **23** Department of Internal Medicine, Flevo Hospital, Almere, Netherlands, **24** Department of Internal Medicine, Radboud Centre for Infectious Diseases, Radboud University Medical Centre, Nijmegen, Netherlands, **25** Department of Internal Medicine and Infectious Diseases, Isala Hospital, Zwolle, Netherlands, **26** Department of Internal Medicine and Infectious Diseases, Maastricht University Medical Centre, Maastricht, Netherlands

☯ These authors contributed equally to this work.
‡ These authors are joint senior authors on this work.
* c.rokx@erasmusmc.nl

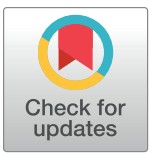

**Data Availability Statement:** Individual participant data that underlie the results reported in this article, after de-identification, will be made available to

researchers who provide a methodologically sound study proposal. Requests for the data on PLWH can be made to the Erasmus MC HIV Eradication Group (EHEG) at eheg@erasmusmc.nl. Contact for inquiries data healthy controls: VACOPID COVID-19 vaccination study: L.P.M. (Leanne) van Leeuwen (l.p.m.vanleeuwen@erasmusmc.nl) and Health Care Workers cohort Erasmus MC: M.C. (Marc) Shamier (m.shamier@erasmsumc.nl).

**Funding:** This trial was funded by The Netherlands Organization for Health Research and Development (ZonMw) (10430072010008 to KB). Control samples were obtained from the VACOPID study, funded by ZonMw (10430072010006 to VASHD and RdDV). DG and RDdV are supported by the Health~Holland grant co-funded by the PPP Allowance made available by the Health~Holland, Top Sector Life Sciences & Health, to stimulate public –private partnerships (EMCLHS20017 to DG and RDdV). https://www.zonmw.nl/en/ The funders had no role in study design, data collection and analysis, decision to publish, or preparation of the manuscript.

**Competing interests:** I have read the journal's policy and the authors of this manuscript have the following competing interests: All authors have completed the ICMJE disclosure form and declare no competing interests exist directly related to the submitted work Conflicts of interest outside the submitted work CR has received research grants from ViiV, Gilead, ZonMW, AIDSfonds, Erasmus MC, and Health~Holland and honorariums for advisory boards from Gilead and ViiV; WFWB declares reimbursement for participation of patient in trial by GSK to institution. DG and RDdV are supported by the Health~Holland grant EMCLHS20017 co-funded by the PPP Allowance made available by the Health~Holland, Top Sector Life Sciences & Health, to stimulate public–private partnerships. RDdV is listed as inventor of the fusion inhibitory lipopeptide [SARSHRC-PEG4]2-chol on a provisional patent application. VASHD has received research grants from ZonMw, Horizon 2020 – Marie Curie-Sklodowska, Takeda and payments for lectures and advisory boards from Takeda, CSL Behring, Pharming and GSK. KCES received honorariums for advisory boards from Gilead and ViiV. BJAR declares research grants from Gilead and MSD and honorary for advisory boards for Astra Zeneca, Roche, Gilead, F2G all outside the context of this work. RvM received consultancies fees paid to their institution from ViiV; Gilead; MSD, received research grants paid to their institution from ViiV; Gilead All other authors declare hat no competing interests exist.

## Abstract

### Background

Vaccines can be less immunogenic in people living with HIV (PLWH), but for SARS-CoV-2 vaccinations this is unknown. In this study we set out to investigate, for the vaccines currently approved in the Netherlands, the immunogenicity and reactogenicity of SARS-CoV-2 vaccinations in PLWH.

### Methods and findings

We conducted a prospective cohort study to examine the immunogenicity of BNT162b2, mRNA-1273, ChAdOx1-S, and Ad26.COV2.S vaccines in adult PLWH without prior COVID-19, and compared to HIV-negative controls. The primary endpoint was the anti-spike SARS-CoV-2 IgG response after mRNA vaccination. Secondary endpoints included the serological response after vector vaccination, anti-SARS-CoV-2 T-cell response, and reactogenicity. Between 14 February and 7 September 2021, 1,154 PLWH (median age 53 [IQR 44–60] years, 85.5% male) and 440 controls (median age 43 [IQR 33–53] years, 28.6% male) were included in the final analysis. Of the PLWH, 884 received BNT162b2, 100 received mRNA-1273, 150 received ChAdOx1-S, and 20 received Ad26.COV2.S. In the group of PLWH, 99% were on antiretroviral therapy, 97.7% were virally suppressed, and the median CD4+ T-cell count was 710 cells/µL (IQR 520–913). Of the controls, 247 received mRNA-1273, 94 received BNT162b2, 26 received ChAdOx1-S, and 73 received Ad26.COV2.S. After mRNA vaccination, geometric mean antibody concentration was 1,418 BAU/mL in PLWH (95% CI 1322–1523), and after adjustment for age, sex, and vaccine type, HIV status remained associated with a decreased response (0.607, 95% CI 0.508–0.725, $p < 0.001$). All controls receiving an mRNA vaccine had an adequate response, defined as >300 BAU/mL, whilst in PLWH this response rate was 93.6%. In PLWH vaccinated with mRNA-based vaccines, higher antibody responses were predicted by CD4+ T-cell count 250–500 cells/µL (2.845, 95% CI 1.876–4.314, $p < 0.001$) or >500 cells/µL (2.936, 95% CI 1.961–4.394, $p < 0.001$), whilst a viral load > 50 copies/mL was associated with a reduced response (0.454, 95% CI 0.286–0.720, $p = 0.001$). Increased IFN-γ, CD4+ T-cell, and CD8+ T-cell responses were observed after stimulation with SARS-CoV-2 spike peptides in ELISpot and activation-induced marker assays, comparable to controls. Reactogenicity was generally mild, without vaccine-related serious adverse events. Due to the control of vaccine provision by the Dutch National Institute for Public Health and the Environment, there were some differences between vaccine groups in the age, sex, and CD4+ T-cell counts of recipients.

### Conclusions

After vaccination with BNT162b2 or mRNA-1273, anti-spike SARS-CoV-2 antibody levels were reduced in PLWH compared to HIV-negative controls. To reach and maintain the same serological responses as HIV-negative controls, additional vaccinations are probably required.

**Abbreviations:** AE, adverse event; AIM, activation-induced marker; BAU, binding antibody units; cART, combination antiretroviral therapy; DMSO, dimethyl sulfoxide; ELISpot, enzyme-linked immune absorbent spot; GMC, geometric mean concentration; MOG, myelin oligodendrocyte glycoprotein; OR, odds ratio; PBMC, peripheral blood mononuclear cell; PLWH, people living with HIV; SAE, serious adverse event; SFC, spot-forming cell; VOC, variant of concern.

## Trial registration

The trial was registered in the Netherlands Trial Register (NL9214). https://www.trialregister.nl/trial/9214.

## Author summary

### Why was this study done?

- The efficacy of SARS-CoV-2 vaccines in people living with HIV (PLWH) is not well characterised.

- HIV has been repeatedly associated with lower immune responses to other vaccines, and this diminished response is strongly correlated with CD4+ T-cell count.

- The SARS-CoV-2 vaccines BNT162b2, mRNA-1273, ChAdOx1-S, and Ad26.COV2.S showed good protection against severe COVID-19 and hospitalisation in phase III registration trials; however, the number of PLWH in these trials was very limited.

### What did the researchers do and find?

- We initiated a nationwide prospective study including 1,154 PLWH and 440 HIV-negative controls.

- We show that lower antibody levels are seen in PLWH compared to controls after completion of the vaccination schedule, regardless of the vaccine received.

- All controls receiving an mRNA vaccine had an adequate response, defined as >300 BAU/mL, whilst in PLWH this response rate was 93.6%. In multivariable analyses, having HIV had the largest negative effect on antibody responses following vaccination, more than both age and sex.

- Following mRNA vaccination, the antibody response was higher in PLWH with CD4+ T-cell counts between 250 and 500 cells/μL or higher than 500 cells/μL (both $p < 0.001$), while those with <250 cells/μL had a lower response. In PLWH, age above 65 years and being born as male were associated with lower antibody concentrations as well (both $p < 0.001$).

### What do these findings mean?

- Clinicians should particularly be aware of potential lower vaccine responses in elderly PLWH and those with lower cellular immunity or evidence of acquired immunodeficiency syndrome.

- PLWH may require additional vaccinations on top of standard regimens to achieve and keep protection against SARS-CoV-2 at similar levels to HIV-negative controls.

- In these participants, with the vaccines studied, mRNA-based vaccine strategies are to be preferred over vector-based ones.

## Introduction

At the end of 2019, severe acute respiratory syndrome coronavirus 2 (SARS-CoV-2) emerged, and the ensuing and ongoing pandemic led to the loss of millions of lives. Highly effective vaccines were quickly developed, and mass vaccination campaigns have become the cornerstone to prevent fatal coronavirus disease (COVID-19) and to quell this pandemic. Four vaccines are currently approved for use in the Netherlands [1–5].

HIV infection is associated with worse COVID-19 outcomes although the underlying mechanism is not yet clear [6]. In most countries, people living with HIV (PLWH) were therefore prioritised for SARS-CoV-2 vaccination. PLWH show diminished responses to a wide variety of vaccines such as hepatitis B and seasonal influenza vaccines compared to HIV-negative individuals [7,8]. This might also hold true for SARS-CoV-2 vaccines. Indicative of potentially lower responses to SARS-CoV-2 vaccines could be that after SARS-CoV-2 infection, lower IgG concentrations and neutralising antibody titres were found in PLWH compared to controls [9]. Data are scarce on SARS-CoV-2 vaccination responses in PLWH; some PLWH were included in the large phase III trials, but data for these participants were not published with the results of these trials [1–5]. Small studies using the ChAdOx1-S vaccine in the UK and South Africa in relatively young PLWH with high CD4+ T-cell counts showed PLWH having comparable responses to controls [10,11]. As for BNT162b2, similar results were shown in a limited number of PLWH [12–14]. The identification of risk factors for a reduced response to SARS-CoV-2 vaccines in PLWH is important as it will help to improve vaccination strategies in PLWH. A good understanding of vaccination response in PLWH is even more important now that variants of concern (VOCs) continue to arise and partially escape vaccine-induced immunity [15], especially considering the possibility of VOCs arising in PLWH unable to clear the virus due to an untreated HIV infection [16].

We hypothesised that SARS-CoV-2 vaccine response in PLWH would be lower than in HIV-negative controls. Our main aim was therefore to investigate the immunogenicity of SARS-CoV-2 vaccinations in PLWH with the vaccines currently approved in the Netherlands —BNT162b2, mRNA-1273, ChAdOx1-S, and Ad26.COV2.S—compared to HIV-negative controls. Additionally, we reviewed the reactogenicity of the vaccines in PLWH.

## Methods

### Study design and participants

We performed a prospective observational cohort study in 22 of the 24 HIV treatment centres in the Netherlands. Participants were recruited via treating physicians or nurses specialised in HIV care. Individuals who were 18 years or older and had a confirmed HIV infection were eligible and were invited for SARS-CoV-2 vaccination by Dutch public health services. Participants with a history of previous SARS-CoV-2 infection demonstrated by PCR or detectable SARS-CoV-2 anti-spike antibodies in serum before vaccination were excluded. Inclusion was stratified according to vaccine type (mRNA or vector), sex assigned at birth, age (18–55, 56–65, or >65 years), and most recent CD4+ T-cell count (<350 and ≥350 cells/µL). In order to recruit a study population that best represented the Dutch population of PLWH, we

continuously monitored recruitment across these strata, and strata were closed for enrolment when a sufficient number had been recruited [17].

Participants received BNT162b2, mRNA-1273, ChAdOx1-S, or Ad26.COV2.S according to manufacturer's regulations as part of the Dutch SARS-CoV-2 vaccination campaign (S1 Text). Vaccination response data from HIV-negative controls were obtained from 2 separate concurrent studies. The first cohort consisted of healthcare workers from the Erasmus University Medical Centre in Rotterdam who were enrolled in a prospective cohort study ($n$ = 385) [18]. Healthcare workers received BNT162b2, mRNA-1273, ChAdOx1-S, or Ad26.COV2.S according to national regulations as described above. The second group consisted of participants who served as non-immunocompromised controls in the Vaccination Against Covid in Primary Immune Deficiencies (VACOPID) study investigating the mRNA-1273 vaccine in people with inborne errors of immunity ($n$ = 55) [19]. They received 2 mRNA-1273 vaccines 4 weeks apart with blood sampling 4 weeks after the second vaccination. None of the controls had a history of COVID-19.

This study is reported as per the Strengthening the Reporting of Observational Studies in Epidemiology (STROBE) guideline (S1 STROBE Checklist).

## Clinical procedures

Between 14 February and 7 September 2021, 1,269 participants were included. Blood samples for serology were collected up to 6 weeks before vaccination in 1,269 participants (pre-vaccination). During study follow-up, 53 participants were excluded after a positive anti-spike antibody test at baseline sampling, and 51 participants were lost to follow-up. Four to six weeks after the completed vaccination schedule, blood draws were performed in 1,165 participants (post-vaccination). In a subgroup of participants willing to participate in extra sampling, additional blood samples were collected for peripheral blood mononuclear cells (PBMCs) at any of the study visits (pre-vaccination, $n$ = 23; post-vaccination, $n$ = 45) or for serology 21 days (±3 days) after the first vaccination (inter-vaccination) ($n$ = 43). Participants were scheduled for longitudinal blood sampling for 2 years for additional analyses, which will be reported separately.

Study variables were collected in an electronic case record file. Study variables that were collected included year of birth, sex assigned at birth (male/female), current use of combination antiretroviral therapy (cART) (yes/no), most recent plasma HIV RNA (copies/mL), most recent CD4+ T-cell count (cells/μL), and nadir CD4+ T-cell count (cells/μL).

Participants received a paper diary or a link to an online questionnaire to record adverse events (AEs) from a predefined list and medication use occurring in the 7 days following each vaccination.

## Laboratory procedures

All serum samples were collected via venepuncture at participating centres. Serum samples before vaccination were analysed at the laboratory of the individual treating centres with the available SARS-CoV-2 antibody test: Wantai SARS-CoV-2 total IgG and IgM ELISAs (Beijing Wantai Biological Pharmacy Enterprise, China), Abbott ARCHITECT SARS-CoV-2 IgG (Abbott Laboratories, Abbott Park, Illinois, US), Siemens Atellica IM SARS-CoV-2 IgG (sCOVG) serology assay (Siemens Healthineers Nederland, The Hague, the Netherlands), or LIAISON by DiaSorin (Saluggia, Italy), depending on local availability and according to the manufacturer's instructions. Serum samples post-vaccination were transported for testing at the Department of Viroscience, Erasmus University Medical Centre, the Netherlands. Binding antibodies against the SARS-CoV-2 spike (S1) were quantified with a validated IgG trimeric

chemiluminescence immunoassay (LIAISON, DiaSorin) with a lower limit of detection at 4.81 binding antibody units (BAU)/mL and a cutoff level for positivity at 33.8 BAU/mL [20].

PBMCs were isolated by density gradient centrifugation (Ficoll-Hypaque, GE Healthcare Life Sciences) and collected in RPMI-1640 (Life Technologies) supplemented with 3% foetal bovine serum (FBS). PBMCs were washed 3 times, frozen in freezing medium (90% FBS, 10% dimethyl sulfoxide [DMSO]), and stored in liquid nitrogen until use.

We used enzyme-linked immune absorbent spot (ELISpot) assays to quantify interferon-γ (INF-γ) secretion in response to SARS-CoV-2 peptides. ELISpot assays were performed on cryopreserved PBMCs using a commercial kit (ImmunoSpot, Cellular Technology). PBMCs were stimulated with peptide pools consisting of SARS-CoV-2 spike protein, SARS-CoV-2 nucleocapsid protein (to exclude PLWH recently infected with SARS-CoV-2), myelin oligo-dendrocyte glycoprotein (MOG) as a negative control, and CEFX (peptide epitopes from different infectious agents) as a positive control (S2 Text). Results are expressed as spot-forming cells (SFCs) per million PBMCs. To exclude nonspecific stimulation of T cells by peptides, specific S responses were calculated by subtracting mean MOG responses from mean spike responses.

T-cell responses were further characterised by activation-induced marker (AIM) assay. PBMCs were incubated with SARS-CoV-2 peptide pools covering the entire spike protein of the WuhanHu1 (wild-type) or B.1.617.2 (Delta) variant. Following stimulation, cells were stained and measured by flow cytometry (FACSLyric, BD Biosciences; S3 Text). SARS-CoV-2-reactive T cells were identified as CD137+OX40+ for CD4+ subtype or CD137+CD69+ for CD8+ subtype. On average, 300,000 cells were measured. The gating strategy can be found in S1 Fig.

## Outcomes

The primary outcome was the magnitude of the anti-spike SARS-CoV-2 IgG response in PLWH 4–6 weeks after the completed vaccination schedule with BNT162b2 or mRNA-1273. This endpoint was chosen because primarily mRNA vaccines were allocated to PLWH in the Netherlands. Secondary outcomes included the antibody response in PLWH after the completed vaccination schedule with ChAdOx1-S or Ad26.COV2.S, and variables associated with the magnitude of antibody level (vaccine type [BNT162b2, mRNA-1273, ChAdOx1-S, or Ad26.COV2.S], sex assigned at birth [male or female], age [birth year 1965–2002, 1955–1964, or 1954 or earlier], nadir CD4+ T-cell count [<250, 250–500, or >500 cells/μL], and most recent CD4+ T-cell count [<250, 250–500, or >500 cells/μL]). Variables associated with hyporesponse and the presence of an antibody response were also analysed (vaccine group [mRNA or vector], sex at birth [male or female], age group [birth year 1965–2002, 1955–1964, or 1954 or earlier], nadir CD4+ T-cell count [<250, 250–500, or >500 cells/μL], and most recent CD4+ T-cell count [<250, 250–500, or >500 cells/μL]). We defined a hyporesponse as lower than 300 BAU/mL, based on previous studies that showed a correlation of antibody concentration of 300 BAU/mL with a neutralising capacity of 1:40 in the wild-type variant [21,22]. The cutoff level for positivity in the DiaSorin assay was 33.8 BAU/mL [20]. In subgroup analyses, anti-spike SARS-CoV-2-specific T-cell response and antibody response 21 days after the first vaccine dose were evaluated. Lastly, we evaluated the tolerability of the administered vaccines by monitoring local and systemic vaccine-related AEs. Severity of reactogenicity was measured as mild (symptoms present but no functional impairment or medication needed), moderate (necessitating medication, no functional impairment), or severe (impairing daily functioning). Serious AEs (SAEs) were assessed for likeliness of association with vaccination by the participating site principal investigator and physician.

## Sample size and statistical analysis plan

When the study started, we did not have confirmed availability of a control group due to the rapid initiation of the national immunisation campaign. We justified the sample size by calculating that 556 PLWH receiving mRNA vaccines would be sufficient to detect, with >80% power, a serological response rate of 90% or lower compared to a hypothetical 95% response rate in controls. When the control group was confirmed, and before the data lock and endpoint analyses, we amended the protocol to update the sample size calculation. Accounting for the imbalance in the number of controls versus PLWH with BNT162b2 and mRNA-1273 vaccinations, we found that 286 controls were sufficient to detect a 20% lower antibody response in PLWH with >80% power and alpha 5%.

Descriptive data are presented as median (interquartile range [IQR]) or $n$ (%). A multivariable linear regression model was used for the analysis of the anti-spike SARS-CoV-2 IgG response. The outcome was transformed using the natural logarithm plus 1 unit: ln(anti-spike SARS-CoV-2 IgG + 1) in order to meet the model assumptions. All participants whose sample was received in the central laboratory for testing who completed the vaccination scheme were included in the analysis (per protocol). The difference between PLWH and controls was captured by the corresponding regression coefficient and 95% confidence interval (CI). The model was further adjusted for differences in vaccine type, age, and sex. A multivariable linear regression model was also used to quantify the difference in ln(anti-spike SARS-CoV-2 IgG + 1) between PLWH and controls for the subset in the sample vaccinated with vector vaccines. A similar model was used to quantify the effect of age, sex, vaccine type, most recent CD4+ T-cell count, and antibody concentration in PLWH. In addition, multivariable logistic regression models were used to calculate odds ratios (ORs) with 95% CIs for the effects of sex, age, nadir CD4+ T-cell count, most recent CD4+ T-cell count, HIV RNA viral load, and vaccine group in PLWH on having a hyporesponse or the presence of a response. In subgroup participants, we evaluated differences from baseline to inter-vaccination and from inter- to post-vaccination, as well as AIM data comparing pre- and post-vaccination time points, by Wilcoxon matched-pairs signed rank test. ELISpot pre- to post-vaccination data and AIM data comparing PLWH and controls were analysed by Mann–Whitney U tests.

Data were analysed using IBM SPSS Statistics 25, R (v. 4.1.2), and GraphPad Prism 8. Flow cytometry data were analysed using FlowJo software version 10.8.1.

## Role of the funding source

The funder of the study had no role in study design, data collection, data analysis, data interpretation, or writing of the report.

## Ethical considerations

The trial was performed in accordance with the principles of the Declaration of Helsinki, Good Clinical Practice guidelines, and the Dutch Medical Research Involving Human Subjects Act (WMO). Written informed consent was obtained from all participants. The trial was reviewed and approved by the Medical Research Ethics Committees United Nieuwegein (MEC-U, reference 20.125). The trial was registered in the Netherlands Trial Register (NL9214).

## Results

### Baseline characteristics

Between 14 February and 7 September 2021, 1,269 PLWH were enrolled (Fig 1). At sampling before vaccination, 53 (4.2%) PLWH had antibodies against SARS-CoV-2 spike protein above

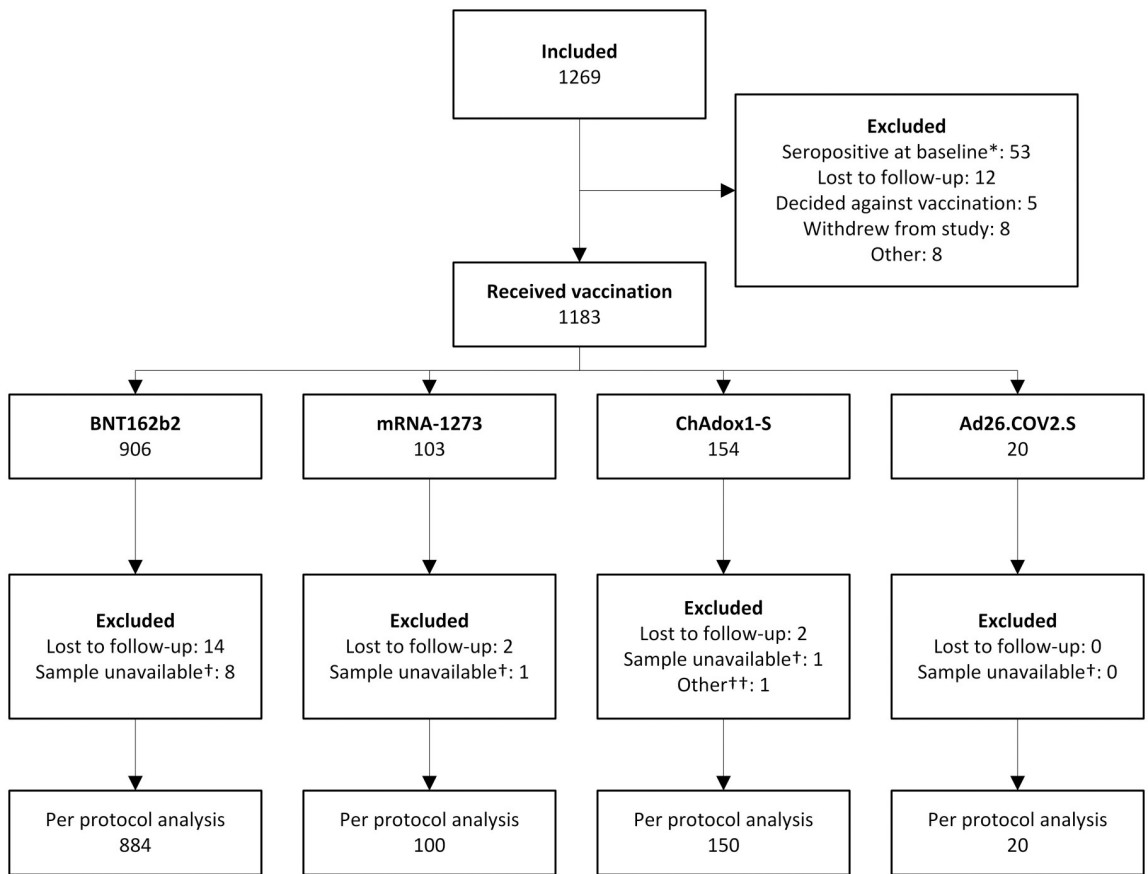

**Fig 1. Flow chart of included PLWH.** *Participants who tested positive for SARS-CoV-2 antibodies in serum at baseline measurement were excluded from further articipation †Samples were not stored adequately or were not sent to be analysed in the central laboratory at the Erasmus University Medical Centre. ††One participant received a combination of ChAdOx1-S and BNT162b2 vaccines and was therefore not included in the per protocol analysis. PLWH, people living with HIV.

the test cutoff and were excluded. Overall, 30 PLWH were lost to follow-up (2.4%), 5 decided against vaccination after inclusion (0.4%), 8 withdrew from the study (0.6%), and 10 samples were not stored adequately or not received in the central laboratory for testing (0.8%). One participant was excluded from the final analysis because they received 2 different vaccines (ChAdOx1-S and BNT162b2). In the final analysis, 76.6% of PLWH received BNT162b2, 8.7% received mRNA-1273, 13.0% received ChAdOx1-S, and 1.7% received Ad26.COV2.S. Included PLWH had a median age of 53 years (IQR 44–60), 85.5% were men, and they had a median CD4+ T-cell count before vaccination of 710 cells/μL (IQR 520–913) (Table 1). The vast majority (99.0%) were on cART and had a suppressed plasma HIV RNA level (97.7% had <50 copies/mL). The control group consisted of 440 people, of whom 94 were vaccinated with BNT162b2 (21.4%), 247 with mRNA-1273 (56.1%), 26 with ChAdOx1-S (5.9%), and 73 with Ad26.COV2.S (16.6%). Their median age was 43 (IQR 33–53), and 28.6% were men. The age distribution across vaccine groups differed, with the majority of PLWH receiving ChAdOx1-S being 56–65 years of age compared to 15%–25% for the other vaccines (S1 Table). Age differences were also seen between the PLWH and control groups, with fewer participants of older ages in the control group. Between the control group and PLWH, there was also a difference in inclusion by sex. All other factors were similar across groups. In the subgroups of PLWH,

**Table 1. Baseline characteristics of HIV-negative controls and PLWH.**

| Characteristic | HIV-negative participants | | | People living with HIV | | |
|---|---|---|---|---|---|---|
| | Overall<br>N = 440 | mRNA vaccines[1]<br>N = 341 (77.5%) | Vector vaccines[2]<br>N = 99 (22.5%) | Overall<br>N = 1,154 | mRNA vaccines[1]<br>N = 984 (85.3%) | Vector vaccines[2]<br>N = 170 (14.7%) |
| **Sex assigned at birth** | | | | | | |
| Male | 126 (28.6%) | 104 (30.5%) | 22 (22.2%) | 987 (85.5%) | 839 (85.2%) | 148 (87.1%) |
| Female | 314 (71.4%) | 237 (69.5%) | 77 (77.8%) | 167 (14.5%) | 145 (14.8%) | 22 (12.9%) |
| **Age category** | | | | | | |
| 18–55 years | 352 (80.0%) | 284 (83.3%) | 68 (68.7%) | 703 (60.9%) | 674 (68.5%) | 29 (17.1%) |
| 56–65 years | 74 (16.8) | 43 (12.6%) | 31 (31.3%) | 291 (25.2%) | 157 (16.0%) | 134 (79.3%) |
| >65 years | 14 (3.2%) | 14 (4.1%) | 0 | 160 (13.9%) | 153 (15.6%) | 7 (4.1%) |
| **On cART** | | | | | | |
| Yes | NA | NA | NA | 1,142 (99.0%) | 972 (98.8%) | 170 (100%) |
| No | NA | NA | NA | 12 (1.0%) | 12 (1.2%) | 0 |
| **Most recent plasma HIV viral load** | | | | | | |
| <50 copies/mL | NA | NA | NA | 1,127 (97.7%) | 960 (97.6%) | 167 (98.2%) |
| ≥50 copies/mL | NA | NA | NA | 27 (2.3%) | 24 (2.4%) | 3 (1.8%) |
| **Most recent CD4+ T-cell count** | | | | | | |
| <250 cells/μL | NA | NA | NA | 41 (3.6%) | 35 (3.6%) | 6 (3.5%) |
| 250–500 cells/μL | NA | NA | NA | 224 (19.4%) | 189 (19.2%) | 35 (20.6%) |
| >500 cells/μL | NA | NA | NA | 889 (77.0%) | 760 (77.2%) | 129 (75.9%) |
| **Nadir CD4+ T-cell count** | | | | | | |
| <250 cells/μL | NA | NA | NA | 443 (38.4%) | 365 (37.1%) | 78 (45.9%) |
| 250–500 cells/μL | NA | NA | NA | 376 (32.6%) | 330 (33.5%) | 46 (27.1%) |
| >500 cells/μL | NA | NA | NA | 152 (13.2%) | 133 (13.5%) | 19 (11.2%) |
| Unknown | NA | NA | NA | 183 (15.9%) | 156 (15.9%) | 27 (15.9%) |
| **Days between doses*** | 28 (25–28) | 28 (25–28) | 56 (56–70) | 35 (35–36) | 35 (35–36) | 70 (49–77) |
| **Days between second[†] vaccination and blood draw** | 29 (27–33) | 28 (25–31) | 30 (28–32) | 30 (28–34) | 30 (28–34) | 30 (28–34) |

Data are n (%) or median (IQR).

[1]BNT162b2 or mRNA-1273.

[2]ChAdOx1-S or Ad26.COV2.S.

*Does not apply for Ad26.COV2.S.

[†]First and only vaccination for Ad26.COV2.S.

cART, combination antiretroviral therapy; IQR, interquartile range; NA, not applicable.

baseline characteristics reflected the characteristics of included PLWH, with most participants receiving the BNT162b2 vaccine (66.0%–95.3%) and being male (76.6%–85.7%) (S2 Table).

## Humoral responses

In all vaccines investigated, after completion of the vaccination schedule, antibody concentrations were lower in PLWH compared to controls (Fig 2). In participants vaccinated with an mRNA vaccine, the geometric mean concentration (GMC) was 1,418 BAU/mL in PLWH (95% CI 1,322–1,523) and 3,560 BAU/mL in controls (95% CI 3,301–3,840). All controls receiving an mRNA vaccine had an adequate response, defined as >300 BAU/mL, whilst in PLWH this response rate was 93.6%. Nine PLWH had a response below the limit of detection. With regard to the primary endpoint, after adjusting for age, vaccine, and sex, HIV infection remained associated with a 39.35% lower antibody concentration in PLWH compared to HIV-negative controls receiving an mRNA vaccine (0.607, 95% CI 0.508–0.725, $p < 0.001$) (S3

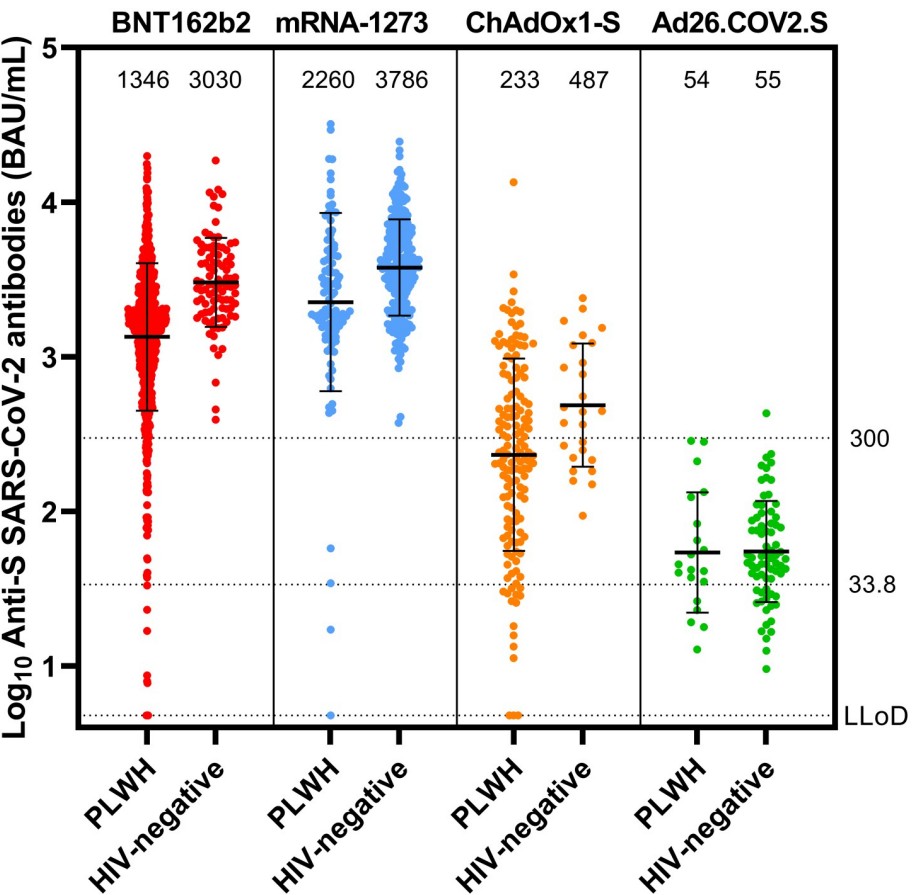

**Fig 2. Antibody concentration in PLWH and controls after vaccination.** Anti-spike SARS-CoV-2 binding antibodies after BNT162b2 (HIV-negative, $n$ = 94; PLWH, $n$ = 884), mRNA-1273 (HIV-negative, $n$ = 247; PLWH, $n$ = 100), ChAdOx1-S (HIV-negative, $n$ = 26; PLWH, $n$ = 150), or Ad26.COV2.S (HIV-negative, $n$ = 73; PLWH, $n$ = 20) vaccination in PLWH and the HIV-negative control group. The thick horizontal bar shows the geometric mean concentration, also indicated in the numbers above the graphs, with error bars showing geometric standard deviation. The dotted lines show the lower limit of detection of the performed test (4.81 BAU/mL), the positivity cutoff (33.8 BAU/mL), and the hyporesponse cutoff (300 BAU/mL). BAU, binding antibody units; LLoD, lower limit of detection; PLWH, people living with HIV; S, spike.

Table). The estimated effect of having an HIV infection was larger than the estimated effects of male sex (23.05% [0.769, 95% CI 0.667–0.888]) and age over 65 years (35.47% [0.645, 95% CI 0.544–0.765]), which were both significantly associated with a worse vaccine response ($p <$ 0.001 for both).

Regarding factors possibly related to antibody responses in PLWH who received mRNA vaccines, receiving mRNA-1273 was associated with a 57.15% higher response than receiving BNT162b2 (1.572, 95% CI 1.225–2.018, $p <$ 0.001) (S4 Table). Male sex (0.693, 95% CI 0.555–0.865, $p$ = 0.001) and age over 65 years (0.654, 95% CI 0.526–0.814, $p <$ 0.001) were associated with lower responses of 30.72%, and 34.56%, respectively. We found no association between nadir CD4+ T-cell count and antibody responses, but found a significant effect of 54.62% lower antibodies when the HIV RNA level was over 50 copies/mL (0.454, 95% CI 0.286–0.720, $p$ = 0.001). The largest effect on antibody levels was associated with having a current CD4+ T-cell count between 250 and 500 cells/μL (2.845, 95% CI 1.876–4.314) or over 500 cells/μL (2.936, 95% CI 1.961–4.394) (both $p <$ 0.001), with increased antibody concentrations

of 184.34% and 193.59%, respectively, compared to PLWH with CD4+ T-cell count under 250 cells/μL.

In participants receiving vector vaccines, after adjustment, HIV was significantly associated with a 39.47% lower antibody response, comparable to that for mRNA vaccines (0.605, 95% CI 0.387–0.945, $p = 0.027$) (S5 Table). Within this group of PLWH receiving vector vaccines, unlike mRNA vaccines, age, sex, and a detectable viral load were not associated with antibody responses, but receiving Ad26.COV2.S was associated with a 87.67% lower response (0.123, 95% CI 0.051–0.300, $p < 0.001$) and having a recent CD4+ T-cell count of 250 to 500 cells/μL or over 500 cells/μL was associated with a better response (both $p < 0.001$) (S6 Table).

Hyporesponse percentages, as well as the percentages of participants with no response, were higher for every vaccine in PLWH compared to controls (S1 Table). In PLWH, after adjustment, receiving a vector vaccine ($p < 0.001$, OR 0.036), age over 65 years ($p < 0.001$, OR 0.282), and viral load over 50 copies/mL ($p = 0.017$, OR 0.266) were associated with an antibody response under 300 BAU/mL (S7 Table). Having a most recent CD4+ T-cell count between 250 and 500 cells/μL or over 500 cells/μL was significantly associated with an antibody response of 300 BAU/mL or higher (both $p < 0.001$, OR 8.143 and 9.177, respectively). Sex and nadir CD4+ T-cell count were not associated with hyporesponse. When looking at the presence of a response in PLWH, receiving a vector vaccine ($p < 0.001$, OR 0.029) was associated with having no antibody response (<33.8 BAU/mL) (S8 Table). Being 56 to 65 years of age ($p = 0.009$, OR 3.697) and having most recent CD4+ T-cell count between 250 and 500 cells/μL or over 500 cells/μL (both $p < 0.001$, OR 7.573 and 16.894, respectively) were significantly associated with an antibody response of ≥33.8 BAU/mL. Sex, viral load, and nadir CD4+ T-cell count were not associated with having no antibody response.

In the subgroup of PLWH in whom extra sampling was performed 21 days after the first vaccination ($n = 43$, 95.3% with BNT162b2), we found a response rate of 83.3%, with a GMC of 148 BAU/mL in between vaccinations and 1,952 BAU/mL after vaccinations (S2 Fig).

## Cellular responses

In the ELISpot assay, spike-specific T-cell responses measured as IFN-γ after deduction of MOG, increased from a median of 27.5 SFCs per million PBMCs before vaccination to 152.5 SFCs per million PBMCs post-vaccination ($p = 0.002$) (Fig 3A). mRNA and vector vaccine responses are shown separately in S3A and S3B Fig, and the responses after subtraction of DMSO with medium control are shown in S3C Fig. In PLWH, stimulating PBMCs with the negative control peptide pool MOG already induced a IFN-γ response, resulting in high background (S4 Fig). In order to determine whether any participants had SARS-CoV-2 infections that were missed via anti-spike SARS-CoV-2 antibody testing, nucleocapsid was added to the assay. No responses for nucleocapsid were seen after subtraction of MOG. Additionally, CD4+ and CD8+ T-cell responses were assessed in an AIM assay. SARS-CoV-2-specific CD4+ T-cell (CD137+OX40+) and CD8+ T-cell responses (CD137+CD69+) both increased compared to baseline following vaccination and after correction for DMSO ($p = 0.005$ and $p = 0.008$, respectively) (Fig 3B and 3C, respectively). CD4+ and CD8+ T-cell responses in PLWH against the Delta variant were similar before and after vaccination. Importantly, both CD4+ and CD8+ T-cell responses were of similar magnitude between PLWH and HIV-negative individuals.

## Reactogenicity

In PLWH, the questionnaire to record AEs and medication use was completed 1,039 (90.0%) times after the first vaccination and 1,026 (90.4%) times after the second vaccination. Overall,

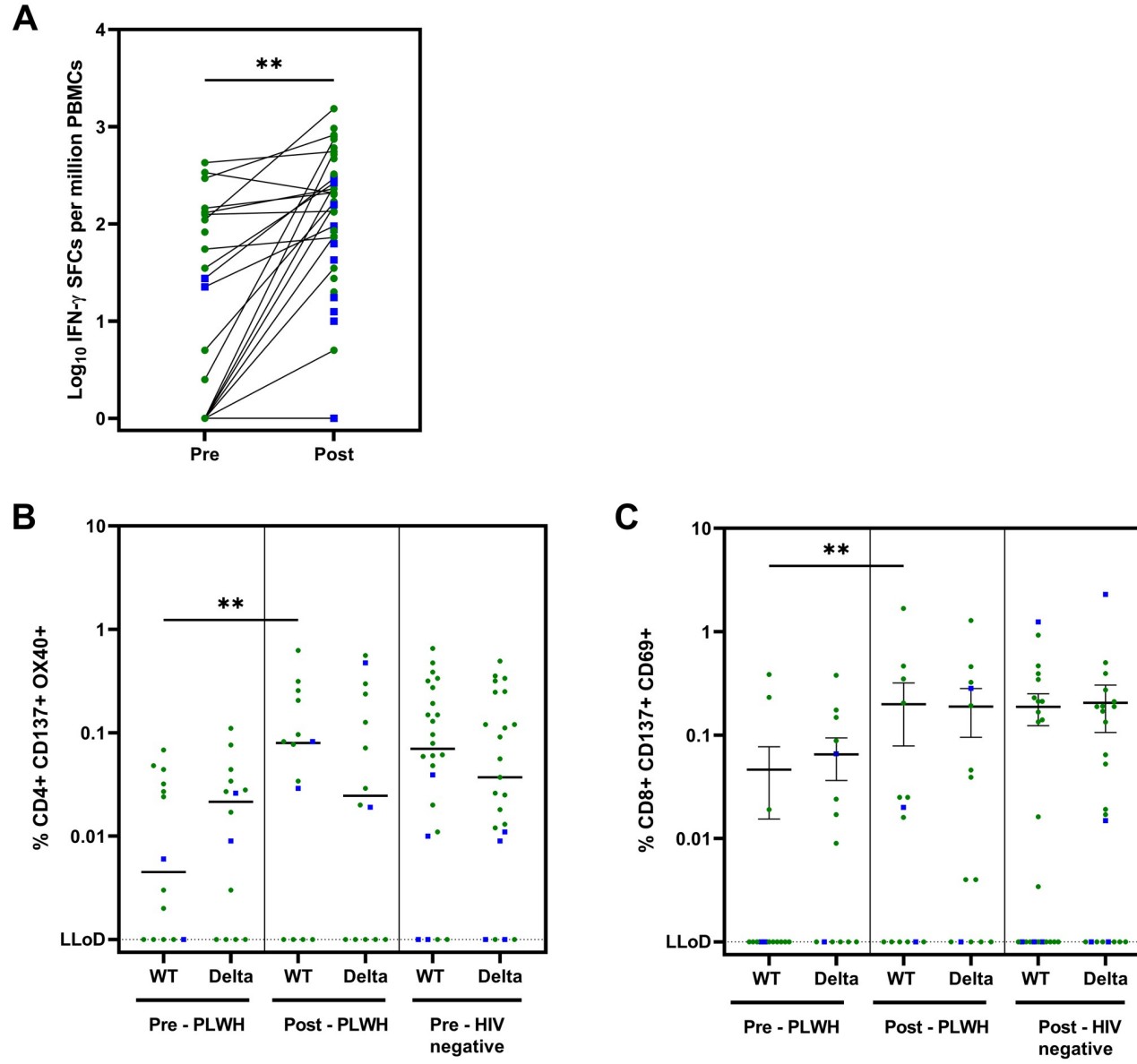

**Fig 3. Cellular immune responses against SARS-CoV-2 in subgroup participants (PLWH).** (A) Cellular immune response to wild-type spike by ELISpot assay (Pre, $n = 23$; Post, $n = 45$): IFN-γ SFCs after subtraction of MOG. Statistics performed using Mann–Whitney U test: $p = 0.002$. Negative responses: Pre, 8; Post, 5. (B) Cellular immune response to wild-type and Delta spike in AIM assay ($n = 14$): percentage of CD4+ CD137+ OX40+ T-cells after subtraction of DMSO. Dotted line shows the LLoD at 0.001%. Statistics between pre- and post-vaccination for WT in PLWH performed using Wilcoxon matched-pairs signed rank test ($p = 0.005$); pre- and post-vaccination for Delta: ns. Statistics between PLWH and controls with Mann–Whitney U test: not significant. (C) Cellular immune response to wild-type and Delta spike in AIM assay ($n = 14$): percentage of CD8+ CD137+ CD69 + T-cells after subtraction of DMSO. Dotted line shows the LLoD at 0.001%. Statistics performed using Wilcoxon matched-pairs signed rank test ($p = 0.008$); pre- and post-vaccination for Delta: ns. Statistics between PLWH and controls with Mann–Whitney U test: not significant. Green circles: mRNA vaccines; blue squares: vector-based vaccines. Pre: before vaccination; Post: 4–6 weeks after second vaccination. AIM, activation-induced marker; BAU, binding antibody units; cART, combination antiretroviral therapy; DMSO, dimethyl sulfoxide; ELISpot, enzyme-linked immune absorbent spot; GMC, geometric mean concentration; INF-γ, interferon-γ; LLoD, lower limit of detection; MOG, myelin oligodendrocyte glycoprotein; PBMC, peripheral blood mononuclear cell; PLWH, people living with HIV; SFC, spot-forming cell; WT, wild type.

**Table 2. Reactogenicity in PLWH.**

| Adverse event | Overall N = 2,065 | BNT162b2 N = 1,593 (77.1%) | | mRNA-1273 N = 177 (8.6%) | | ChAdOx1-S N = 276 (13.4%) | | Ad26.COV2.S N = 19 (0.9%) |
|---|---|---|---|---|---|---|---|---|
| | | 1st dose n = 791 (49.7%) | 2nd dose n = 802 (50.3%) | 1st dose n = 90 (50.8%) | 2nd dose n = 87 (49.2%) | 1st dose n = 139 (50.4%) | 2nd dose n = 137 (49.6%) | |
| **Any AE** | 1,083 (52.4%) | 437 (55.2%) | 358 (48.0%) | 56 (62.2%) | 61 (70.1%) | 72 (51.8%) | 63 (46.0%) | 9 (47.4%) |
| **Local AE** | | | | | | | | |
| Pain at the injection site | 917 (44.4%) | 393 (49.7%) | 325 (40.5%) | 52 (57.8%) | 57 (65.5%) | 51 (36.7%) | 32 (23.4%) | 7 (36.8%) |
| Redness at the injection site | 83 (4.0%) | 27 (3.4%) | 29 (3.6%) | 5 (5.6%) | 7 (8.0%) | 7 (5.0%) | 8 (5.8%) | 0 |
| **Systemic AE** | | | | | | | | |
| Generalised myalgia | 269 (13.0%) | 84 (10.6%) | 107 (13.3%) | 15 (16.7%) | 26 (29.9%) | 21 (15.1%) | 13 (9.5%) | 3 (15.8%) |
| Fever | 72 (3.5%) | 14 (1.8%) | 26 (3.2%) | 4 (4.4%) | 15 (17.2%) | 8 (5.8%) | 5 (3.6%) | 0 |
| Headache | 377 (18.3%) | 126 (15.9%) | 134 (16.7%) | 20 (22.2%) | 28 (32.2%) | 41 (29.5%) | 24 (17.5%) | 4 (21.1%) |
| Rash other than injection site | 19 (0.9%) | 12 (1.5%) | 2 (0.2%) | 1 (1.1%) | 3 (3.4%) | 0 | 0 | 1 (5.3%) |
| Lymphadenopathy | 41 (2.0%) | 13 (1.6%) | 18 (2.2%) | 2 (2.2%) | 5 (5.7%) | 0 | 3 (2.2%) | 0 |
| **Medication use** | | | | | | | | |
| Any medication | 346 (16.8%) | 116 (14.7%) | 120 (15.0%) | 19 (21.1%) | 26 (29.9%) | 34 (24.5%) | 23 (16.8%) | 2 (10.5%) |
| Paracetamol | 281 (81.2%) | 95 (81.9%) | 97 (80.8%) | 16 (84.2%) | 22 (84.6%) | 30 (88.2%) | 21 (91.3%) | 2 (100%) |
| NSAID | 34 (9.8%) | 13 (11.2%) | 13 (10.8%) | 3 (15.8%) | 4 (15.4%) | 2 (5.9%) | 1 (4.3%) | 0 |
| Other | 31 (9.0%) | 8 (6.9%) | 10 (8.3%) | 0 | 0 | 2 (5.9%) | 1 (4.3%) | 0 |

Reactogenicity after vaccine administration occurring within 7 days after each dose in PLWH. Data are *n* (%). AE, adverse event; NSAID, non-steroidal anti-inflammatory drug; PLWH, people living with HIV.

more than half (52.4%) of the participants reported any AE (Table 2). For those who received 2 doses, the frequency of AEs did not increase after the second vaccination (first dose, 55.2%; second dose, 49.6%). The percentage of participants reporting AEs after the first and second vaccination, respectively, was 55.2% and 48.0% for BNT162b2, 62.2% and 70.1% for mRNA-1273, and 51.8% and 46.0% for ChAdOx1-S, and after the single Ad26.COV2.S vaccination was 47.4%. The most reported local reaction was pain at the injection site (44.4%). The most common systemic reactions were myalgia (13.0%) and headache (18.3%). When AEs occurred, most were mild (1,159, 65.2%) or moderate (523, 29.4%) in severity and self-limiting (Fig 4). Analgesic or antipyretic drug use was necessary in 346 (16.8%) of all participants, for a cumulative 769 AEs, of which 400 (52.0%) were moderate and 57 (7.4%) were severe AEs. Paracetamol (81.2%) was most commonly used. Ten SAEs were reported, and all were considered unrelated to vaccination. One participant visited the emergency department 3 days after vaccination with pain in the arm and chest and shortness of breath, in whom a pulmonary embolism was excluded and symptoms resolved. Two participants were admitted for a chronic obstructive pulmonary disease exacerbation. The 7 other SAEs were elective heart surgery, intestinal perforation after infection, *Campylobacter jejuni* infection, bicycle accident, pyloric stenosis, hip fracture after a fall, and death due to a cardiac arrest. There was no discontinuation of the vaccination series due to vaccine-related AEs.

## Discussion

Limited data exist on SARS-CoV-2 vaccination responses in PLWH. Here, we show that mRNA induces lower SARS-CoV-2 S1-specific IgG levels in PLWH compared to controls when measured with the LIAISON assay, even after correction for age, sex, and vaccine type.

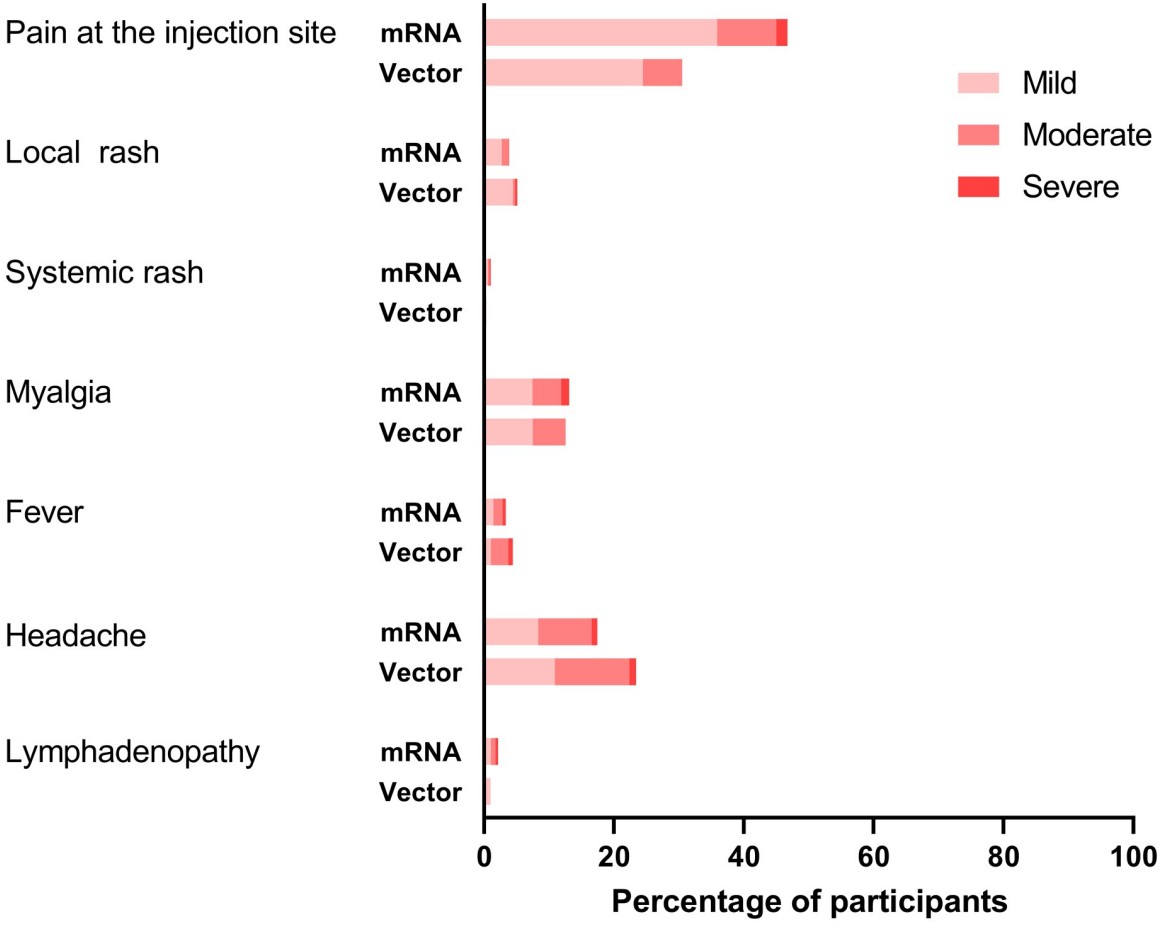

**Fig 4. Severity of adverse events.** Severity of adverse events present in the 7 days after vaccination in people living with HIV, comparing mRNA (BNT162b2 and mRNA-1273) and vector (ChAdOx1-S and Ad26.COV2.S) vaccines, shown as percentage (%) of participants.

PLWH receiving a vector vaccine, of an older age, and with lower CD4+ T-cell counts have more impaired antibody responses. As expected, the SARS-CoV-2 vaccines were well tolerated in PLWH, without vaccine-related discontinuations or SAEs.

Our primary result analysis in participants receiving mRNA vaccines contrasts with most of the small cohort studies performed in PLWH where the authors found similar responses as in controls [12–14]. With regard to vector vaccines in our study, similar effects were seen as for mRNA vaccines, with lower antibody concentrations in PLWH compared to controls. This differs from previously published results on ChAdOx1-S, where no differences were seen between PLWH and HIV-negative participants [10,11]. Most, if not all, of the previous studies, of both mRNA and vector vaccines, had not been powered to detect a predefined size of impact of HIV on vaccination response. Therefore, this discrepancy is probably a type II error of the previous much smaller studies. Other reasons that may explain why some of the previous studies did not find a lower response in PLWH may be the use of qualitative rather than quantitative antibody responses, showing presence of antibodies rather than magnitude of antibody response. Additionally, these studies included younger participants and a larger percentage of female participants, as well as participants with undetectable viral load and high CD4+ T-cell counts. In contrast to the phase III trials, we observed a lower response in male participants [1,2]. This difference in response between sexes has previously been observed in PLWH,

specifically in a yellow fever vaccination study [23]. Additionally, age is also known to influence the immune response to SARS-CoV-2. Our results confirm this in PLWH, in whom more immunosenescence is seen compared to HIV-negative controls [24,25].

Overall, we found an increase in T-cell responses both for activation of CD4+ and CD8+ T-cells and for cytokine production when exposed to SARS-CoV-2 spike protein. There was still a relevant proportion of PLWH in whom T-cell responses after stimulation could not be measured. However, in the AIM assay, this was also observed in controls. Low response or negative responses have also been seen in ELISpot assays performed previously in healthy participants after vaccination [26]. We observed IFN-γ production in both MOG and the medium with DMSO conditions in the PLWH group. This nonspecific spontaneous IFN-γ production by T cells from PLWH could be due to the higher chronic immune activation and persistent inflammation that has been reported before in PLWH [27]. That indeed immune activation leads to higher background spots in ELISpot assays was previously shown in HIV-negative individuals [28].

AEs occurred in just over half of all cases. When looking at BNT162b2, the overall incidence of AEs did not increase after the second vaccination, although the types of AEs differed somewhat (e.g., systemic events occurred more often after the second dose). Overall, AEs in PLWH were mild and similar to those in phase III trials, both in type and frequency [1–3,5].

This study was performed at 22 of a total 24 HIV treatment sites in the Netherlands. Our recruitment strategy resulted in a large group of PLWH, with reasonable representation of female as well as elderly PLWH and those with lower CD4+ T-cell counts. By stratifying at an early stage, we were able to steer inclusion towards more people of certain groups. Our sample reflected the HIV demographics in the Netherlands, in which the 90-90-90 goals were already reached in 2018 [29].

Several limitations are noteworthy. Because provision of vaccinations was decided by the Dutch National Institute for Public Health and the Environment, we could not fully control the distribution of the available vaccines across age, sex, and CD4+ T-cell strata. Furthermore, there were some differences in age and sex between the PLWH and controls, which we corrected for in our multivariate analyses. Few PLWH with a very low CD4+ T-cell count were enrolled, and even fewer with a viral load > 50 copies/mL. We also cannot fully guarantee that all participants with an antecedent COVID-19 infection were excluded as antibodies may become undetectable over time, but no patients in the subgroup study had measurable responses to nucleocapsid. Unfortunately, only a limited number of PLWH could be included in the subgroup study due to the quick start-up of the study and the limited availability of facilities and people to process the samples. However, the characteristics of the participants in the subgroup receiving the mRNA vaccines are comparable to those of the larger group, and we believe to have included enough participants to detect major clinically relevant signals. Finally, we did not perform neutralisation assays. Whilst neutralisation has been shown to correlate with protection against symptomatic infections of SARS-CoV-2, anti-SARS-CoV-2 RBD IgG concentration was shown to strongly correlate with a surrogate virus neutralisation assay after mRNA vaccination [30,31]. Additionally, in a cohort of healthcare workers, spike-specific IgG antibodies strongly correlated with neutralising antibodies [18,32]. In PLWH, neutralisation titres against the Asp614Gly wild-type strain correlated with antibody responses after vaccination with ChAdOx1-S [11].

The COVID-19 landscape continues to change rapidly as new VOCs are emerging. Recent studies have shown immune escape of the Omicron variant from humoral immunity induced by infection as well as vaccination [26]. However, at least in HIV-negative people, an additional vaccine can boost the immune system and restore antibody cross-neutralisation of the Omicron variant [18]. Additionally, higher antibody levels are associated with greater

protection against symptomatic disease [33,34]. This highlights the important role that additional vaccinations can play in controlling the pandemic. In these patients, with the vaccines studied, mRNA-based vaccine strategies are to be preferred over vector-based ones. Based on the results of this study, we decided to give all participating PLWH with an antibody response below 300 BAU/mL the opportunity to receive an additional mRNA-1273 vaccination [35]. Furthermore, given the safety of the mRNA vaccines, the overall lower vaccine-inducible antibody response observed in PLWH, the observed waning of serum antibody levels over time, and immune escape by VOCs, we think that providing additional vaccinations to all PLWH may optimise protection. Some recent studies confirm that a third dose in PLWH is beneficial and important in light of more frequent breakthrough infections in PLWH after vaccinations [36,37]. Based on these results, an argument can be made for prioritisation and use of a more targeted approach in, for example, older PLWH, in those with lower CD4+ T-cell counts, or based on measured antibody responses and neutralisation capacity after vaccination. Whilst this argument can be made in the case of resource-limited settings, or when prioritisation of vaccination is a requirement, we do not believe this is necessary when vaccinations are freely available. Additionally, this is not a practical approach in the case of limited availability of human resources or of SARS-CoV-2 antibody assays, or where CD4+ T-cell counts are not easily performed.

In conclusion, vaccination of PLWH against SARS-CoV-2 resulted in a lower antibody response compared to HIV-negative controls. Additional vaccinations may therefore be required in order to compensate for this reduced antibody response.

## Supporting information

**S1 Fig. Gating strategy for flow cytometry in AIM assay.**
(DOCX)

**S2 Fig. Serological responses after vaccination in PLWH.**
(DOCX)

**S3 Fig. Cellular immune responses against SARS-CoV-2 in subgroup participants (PLWH).**
(DOCX)

**S4 Fig. ELISpot results in subgroup participants (PLWH).**
(DOCX)

**S1 STROBE Checklist.**
(DOCX)

**S1 Table. Baseline characteristics of HIV-negative participants and PLWH by vaccine.**
(DOCX)

**S2 Table. Subgroup patient characteristics.**
(DOCX)

**S3 Table. Regression model to investigate the difference in antibody concentration between PLWH and HIV-uninfected controls vaccinated with 1 of the 2 available mRNA vaccines (BNT162b2 or mRNA-1273).**
(DOCX)

**S4 Table. HIV-related and HIV-unrelated factors associated with the height of antibody response after vaccination with 1 of the 2 available mRNA vaccines (BNT162b2 or mRNA-**

**1273) in PLWH.**
(DOCX)

**S5 Table. Regression model to investigate the difference in antibody concentration between PLWH and HIV-uninfected controls vaccinated with 1 of the 2 vector vaccines (ChAdOx1-S or Ad26.COV2.S).**
(DOCX)

**S6 Table. HIV-related and HIV-unrelated factors associated with the height of antibody response after vaccination with 1 of the 2 vector vaccines (ChAdOx1-S or Ad26.COV2.S) in PLWH.**
(DOCX)

**S7 Table. Linear regression model to investigate factors associated with the antibody response after completion of the vaccination schedule in PLWH with antibody concentration above the minimal level of clinical protection ($\geq$300 BAU/mL).**
(DOCX)

**S8 Table. Linear regression model to investigate factors associated with the antibody response after completion of the vaccination schedule in PLWH with a quantifiable antibody concentration ($\geq$33.8 BAU/mL).**
(DOCX)

**S1 Text. Additional information on study design and participants.**
(DOCX)

**S2 Text. Additional information on ELISpot assay.**
(DOCX)

**S3 Text. Additional information on AIM assay.**
(DOCX)

## Acknowledgments

Foremost, we would like to thank all participants of the study for helping advance science. We also want to thank the following people who helped in the recruitment of participants: Aniek Adams, A. Boonstra, Marjolein van Broekhuizen, Margo van der Burg-van der Plas, A. Cents-Bosma, Willemien Dorama, T. Duijf, S. Faber, Natasja van Holten, Astrid van Hulzen, L. M. Kampschreur, Annemarie van der Kraan, Inge de Kroon, Laura Laan, Eliane Leyten, Vera Maas, P. A. der Meulen, Femke Mollema, Suzanne de Munnik, Hans-Erik Nobel, Vincent Peters, Simone Phaf, M. Pietersma, Frank Pijnappel, Leontine M. M. van der Prijt, Linda Scheiberlich, Jasmijn Steiner, Jolanda M. van der Swaluw, Maartje Wagemaker, Annouschka Weijsenveld, Marc van Wijk, Sieds Wildenbeest, and Sabine van Winden. We would like to thank all our colleague internist–infectious disease specialists in the Netherlands who helped with the patient recruitment. Finally, we would like to thank Alessandro Sette and Alba Grifoni (La Jolla Institute for Immunology, La Jolla, San Diego, US) for providing the peptide pools used in the AIM assay.

## Author Contributions

**Conceptualization:** Kathryn S. Hensley, Daryl Geers, Corine H. GeurtsvanKessel, Yvonne M. Mueller, Alicja Gorska, Marc van der Valk, Rory D. de Vries, Peter D. Katsikis, Bart J. A. Rijnders, Kees Brinkman, Anna H. E. Roukens, Casper Rokx.

**Data curation:** Kathryn S. Hensley, Daryl Geers, Bart J. A. Rijnders, Kees Brinkman, Anna H. E. Roukens, Casper Rokx.

**Formal analysis:** Kathryn S. Hensley, Daryl Geers, Grigorios Papageorgiou.

**Funding acquisition:** Rory D. de Vries, Bart J. A. Rijnders, Kees Brinkman, Anna H. E. Roukens, Casper Rokx.

**Investigation:** Kathryn S. Hensley, Marlou J. Jongkees, Daryl Geers, Virgil A. S. H. Dalm, Hanka Steggink, Alicja Gorska, Susanne Bogers, Jan G. den Hollander, Wouter F. W. Bierman, Luc B. S. Gelinck, Emile F. Schippers, Heidi S. M. Ammerlaan, Marc van der Valk, Corine E. Delsing, Elisabeth H. Gisolf, Anke H. W. Bruns, Fanny N. Lauw, Marvin A. H. Berrevoets, Kim C. E. Sigaloff, Robert Soetekouw, Judith Branger, Quirijn de Mast, Adriana J. J. Lammers, Selwyn H. Lowe, Bart J. A. Rijnders, Kees Brinkman, Anna H. E. Roukens, Casper Rokx.

**Methodology:** Kathryn S. Hensley, Daryl Geers, Corine H. GeurtsvanKessel, Yvonne M. Mueller, Grigorios Papageorgiou, Rory D. de Vries, Peter D. Katsikis, Bart J. A. Rijnders, Kees Brinkman, Anna H. E. Roukens, Casper Rokx.

**Project administration:** Kathryn S. Hensley, Hanka Steggink, Bart J. A. Rijnders, Kees Brinkman, Anna H. E. Roukens, Casper Rokx.

**Supervision:** Yvonne M. Mueller, Rory D. de Vries, Bart J. A. Rijnders, Kees Brinkman, Anna H. E. Roukens, Casper Rokx.

**Validation:** Kathryn S. Hensley, Daryl Geers, Yvonne M. Mueller, Rory D. de Vries, Bart J. A. Rijnders, Kees Brinkman, Anna H. E. Roukens, Casper Rokx.

**Visualization:** Kathryn S. Hensley, Daryl Geers, Bart J. A. Rijnders, Kees Brinkman, Anna H. E. Roukens, Casper Rokx.

**Writing – original draft:** Kathryn S. Hensley, Kees Brinkman, Anna H. E. Roukens, Casper Rokx.

**Writing – review & editing:** Kathryn S. Hensley, Marlou J. Jongkees, Daryl Geers, Corine H. GeurtsvanKessel, Yvonne M. Mueller, Virgil A. S. H. Dalm, Grigorios Papageorgiou, Hanka Steggink, Alicja Gorska, Susanne Bogers, Jan G. den Hollander, Wouter F. W. Bierman, Luc B. S. Gelinck, Emile F. Schippers, Heidi S. M. Ammerlaan, Marc van der Valk, Marit G. A. van Vonderen, Corine E. Delsing, Elisabeth H. Gisolf, Anke H. W. Bruns, Fanny N. Lauw, Marvin A. H. Berrevoets, Kim C. E. Sigaloff, Robert Soetekouw, Judith Branger, Quirijn de Mast, Adriana J. J. Lammers, Selwyn H. Lowe, Rory D. de Vries, Peter D. Katsikis, Bart J. A. Rijnders, Kees Brinkman, Anna H. E. Roukens, Casper Rokx.

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
