## [Editor Report · Decision Letter 0]

29 Mar 2022

Dear Dr Hensley, 

Thank you for submitting your manuscript entitled "Immunogenicity and reactogenicity of SARS-CoV-2 vaccines in people living with HIV: a nationwide prospective cohort study in the Netherlands" for consideration by PLOS Medicine.

Your manuscript has now been evaluated by the PLOS Medicine editorial staff and I am writing to let you know that we would like to send your submission out for external peer review.

Please re-submit your manuscript within two working days, i.e. by Mar 31 2022 11:59PM.

Kind regards,

Beryne Odeny

PLOS Medicine

---

## [Decision Letter · Decision Letter 1]

1 Jun 2022

Dear Dr. Hensley,

Thank you very much for submitting your manuscript "Immunogenicity and reactogenicity of SARS-CoV-2 vaccines in people living with HIV: a nationwide prospective cohort study in the Netherlands" (PMEDICINE-D-22-01021R1) for consideration at PLOS Medicine. 

[LINK]

In light of these reviews, I am afraid that we will not be able to accept the manuscript for publication in the journal in its current form, but we would like to consider a revised version that addresses the reviewers' and editors' comments. Obviously we cannot make any decision about publication until we have seen the revised manuscript and your response, and we plan to seek re-review by one or more of the reviewers. 

We expect to receive your revised manuscript by Jun 22 2022 11:59PM. Please email us (plosmedicine@plos.org) if you have any questions or concerns.

We look forward to receiving your revised manuscript. 

Sincerely,

Beryne Odeny, 

PLOS Medicine

plosmedicine.org

Comments from Academic editor

I would have preferred to see data presented on neutralizing antibodies as well since that is regarded as the major mediator of protection. However, the authors argument that neutralizing Abs have been shown to correlate with IgG titres is a reasonable point. As an alternative, the authors could consider including data generated using a surrogate ELISA-based neutralization assay which is available commercially and suitable for testing large number of samples without specialised culture facilities. I would also like to have seen a little more analysis of differences in Abs between HIV+ vs HIV-, such as IgG subclasses which are relevant in immunity and can be impacted by HIV.

1) Please revise your title: Please place the study setting before the colon. For example, “… in the Netherlands: a 

nationwide prospective cohort study”

2) The Data Availability Statement (DAS) requires revision. For each data source used in your study: 

3) Abstract:

a) Abstract Background: The final sentence should clearly state the study question.

b) Please ensure that all numbers presented in the abstract are present and identical to numbers presented in the main manuscript text.

c) Please include the actual amounts and/or absolute risk(s) of relevant outcomes

d) Please quantify the main results (please present both 95% CIs and p values).

e) In the last sentence of the Abstract Methods and Findings section, please describe the main limitation(s) of the study's methodology.

4) Author summary - At this stage, we ask that you reformat your non-technical Author Summary. The Author Summary should immediately follow the Abstract in your revised manuscript. This text is subject to editorial change and should be distinct from the scientific abstract. The summary should be accessible to a wide audience that includes both scientists and non-scientists. Please see our author guidelines for more information: https://journals.plos.org/plosmedicine/s/revising-your-manuscript#loc-author-summary.

5) Please conclude the Introduction with a clear description of the study question or hypothesis.

6) Methods:

a) Please report the number of patients and samples, and dates of recruitment

b) Please report how many patients were "lost to follow-up" as used in this study. 

7) Please add the following statement, or similar, to the Methods: "This study is reported as per the Strengthening the Reporting of Observational Studies in Epidemiology (STROBE) guideline (S1 Checklist)."

a) Thank you for providing the STROBE checklist. When completing the checklist, please use section and paragraph numbers, rather than page numbers.

8) Please provide p values in addition to 95% CIs in the main text and tables

9) Please indicate in the figure captions the meaning of the bars and whiskers in figure 2, S3

10) Please use up to 3 decimal points for p-values

11) Please define all abbreviations used in footnotes of tables and figures

12) Please replace "subject" with participant, patient, individual, or person.

13) Please replace “HIV patients” with “people living with HIV”

14) Please re-label the Conclusion section to Discussion. 

15) Please organize the Discussion as follows: a short, clear summary of the article's findings; what the study adds to existing research and where and why the results may differ from previous research; strengths and limitations of the study; implications and next steps for research, clinical practice, and/or public policy; one-paragraph conclusion.

16) References: Please include access dates for all weblinks and ensure that all weblinks are current and accessible e.g ref # 28

17) We suggest you copyedit your manuscript for language usage, spelling, and grammar. 

Comments from the reviewers:

Reviewer #1: This prospective cohort study aims to examine the immunogenicity of BNT162b2, mRNA-1273, ChAdOx1-S and 57 Ad26.COV2.S vaccines in adult PLWH, without prior COVID-19, compared to HIV-negative controls.

Comments:

"Between February-September 2021, 1154 PLWH (median age 53 [IQR 44-60], 86% male) and 440 controls (median age 43 [IQR 33-53], 29% male) were included."

and

"We performed a prospective observational cohort study in 22 of the 24 HIV treatment centres in the Netherlands."

and

"Inclusion was stratified and monitored to best represent represents the Dutch population of PLWH (S1 appendix)"

Can the authors please provide further information on this stratification process? Did the authors consider including any stratification variables within the analysis?

Additionally, can the authors comment on whether the stratification and monitoring they completed was successful (i.e. do the authors consider the analysed sample to be representative of the wider population)?

"Blood samples were collected up to six weeks before vaccination (T0) and four to six weeks after the completed vaccination schedule (T2)."

Did the authors consider including time to sample collection (after completed vaccination schedule) as a covariate in the analysis?

"In a subgroup, additional blood samples were collected, 21 days (+/- three days) after the first vaccination for serology (T1) or peripheral blood mononuclear cells (PBMCs) at any of the study visits."

Can the authors please provide further information here? For instance, can the authors please specify in the Methods the number of participants included in these subgroups, and how the participants were selected (e.g. randomly)?

"ELISpot T0 to T2 data and AIM data comparing PLWH and controls were analysed by Mann-Whitney-U tests."

S2 table shows these subgroups, and it can be seen that there are no HIV negative controls over 65 years of age. Did the authors consider including covariates in this analysis?

Can the authors please discuss the small sample sizes included for these subgroup analyses?

"We justified the sample size by calculating that 556 PLWH receiving mRNA vaccines would be sufficient to detect a serological response rate of 90% or lower compared to a hypothetical 95% response rate in controls with >80% power. When the control group was confirmed, and before the data lock and endpoint analyses, we amended the protocol to update the sample size calculation. Accounting for the imbalance in the number of controls versus PLWH with BNT162b2 and mRNA-1273 vaccinations, we found that 286 controls were sufficient to detect a 20% lower antibody response in PLWH with >80% power and alpha 5%."

The authors suitably provide the basis of the sample size calculations, and associated assessment of study power.

"Median (interquartile range (IQR)), or n (%) for descriptive data were used. A multivariable linear regression model was used for the analysis of the anti-spike SARS-CoV-2 IgG. The outcome was transformed using the natural logarithm plus one unit: ln(anti-spike SARS-CoV-2 IgG+1) in order to meet the model assumptions. "

and

"In subgroup participants we evaluated differences from baseline to T1 and T1 to T2, as well as AIM data comparing T0 and T2 time points, by Wilcoxon matched-pairs signed rank test."

Technically appropriate statistical techniques and modelling methods have been applied by the authors within the context of this research.

"The model was further adjusted for differences in vaccine type, age, and sex. A multivariable linear regression model was also used to quantify the difference in ln(anti-spike SARS-CoV-2 IgG+1) between PLWH versus controls for the subset in the sample vaccinated with vector vaccines. A similar model was used to quantify the effect of age, sex, vaccine type, most recent CD4+ T-cell count, and antibody concentration in PLWH. In addition, multivariable logistic regression models were used to calculate odds ratios with 95%CI for the effects of sex, age, nadir CD4+ T-cell count, most recent CD4+ T-cell count, HIV-RNA viral load, and vaccine group in PLWH on having a hyporesponse or a non response."

Do the authors have information on ethnicity, BMI, or comorbidities that they can consider including in these models?

Table 1: Did the authors consider treating age as a continuous variable?

Did the authors consider including days between doses as a covariate in the models?

Can the authors also please present the fully adjusted models in the tables and figures?

Reviewer #2: Hensley et al. have performed a large-scale study of people living with HIV (PLWH) and their response to a variety of covid19 vaccines. The size of the study is a major strength. The clearest findings from the manuscript are a lower binding antibody response in PLWH compared to controls and a large alteration in binding titres dependent on vaccine type. They do not evaluate antibody neutralisation activity (understandably given the number of samples) nor attempt to relate their binding data to previous studies exploring the relationship between binding/neutralisation titres and vaccine efficacy. Greater discussion of how these binding data fit in the wider cannon of Covid19 vaccine studies (including but not limited to those with PLWH) is needed to make this manuscript of publication quality, please see individual critiques.

The authors do assess cellular immune responses in a sub-group of individuals. However, this data is not particularly clear as the negative control stimulation (an MS associated peptide pool) appears to give a very high response in PLWH. This portion of the data as it currently stands is very confusing and not particularly informative, please see queries below.

Individual critiques

Line 58: The authors state that "the primary endpoint of their study was the anti-spike SARS-CoV-2 IgG response after mRNA vaccination". This is perfectly reasonable, but please acknowledge clearly in the manuscript that neutralisation titres, not just total IgG binding titre, are likely important for vaccine efficacy and that neutralisation hasn't been measured here

Line 67-67: Please make it clear if any participants don't seroconvert and what % this is.

Line 76-77: "To reach and maintain the same serological responses and vaccine efficacy as HIV-negative controls, additional vaccinations are probably required."

This statement needs to be properly caveated as this study has not directly addressed vaccine efficacy nor explored links between the titres measured/serological differences observed and vaccine efficacy.

Line 87-89: "After the SARS-CoV-2 vaccine registration trials, vaccinations were rolled out globally. However, people

with immune deficiencies were only sporadically included in the original phase three SARS-CoV-2 vaccination trials."

Please add details of PLWH included - numbers / % as appropriate

Line 89-91: "Subsequent reports indicated markedly diminished responses in solid organ transplant recipients, and lower responses in haemodialysis patients, stem cell recipients or patient groups on specific immunosuppressant drugs for immune disorders [6-9]."

Please include the magnitudes of these lower responses to put your findings here into context.

Line 97-98: "that after SARS-CoV-2 infection, lower IgG concentrations and neutralising antibody titres were found in PLWH compared to controls [13]"

There are other studies where this was not found, and these should also be cited.

Line 103-4: "A good understanding of vaccination response in PLWH becomes even more important now that variants of concern (VOC) continue to arise and partially escape vaccine induced immunity [19]."

Should be mentioned that persistent infection in PLWH is also a plausible mechanism for generation of VOC (there are at least 2 papers/pre-prints from South African groups on this)

Line 138: "validated IgG Trimeric chemiluminescence immunoassay (DiaSorin Liaison)"

Please provide information on how this assay performs with either of the WHO standards or how results can be converted?

Line 147: Please define MOG at first use.

Line 155 & 159: Please change height to magnitude.

Line 161: "hyporesponse as 50-300 BAU/mL and non-response as <50 BAU/mL, based on previous studies"

If non-response equates to seronegative, please use that term.

Line 204: "At sampling before vaccination, 53 (4.2%) PLWH had antibodies against SARS-CoV-2 above test cut-off and were excluded."

Please state if these were antibodies against N or S?

Line 234: "In all vaccines investigated, antibody concentrations were lower in PLWH compared to controls (Fig 2)."

Please state this after 2 vaccine doses

Fig 2: Antibody concentration in PLWH and controls after vaccination

The results in this figure are intriguing and given the high numbers of samples tested please comment on whether this study is adequately powered to say which vaccine PLWH should have given the defect observed between vaccine types.

Line 254: "We found no association between nadir CD4+ T-cell count and antibody responses"

Since Ab response in PLWH have such a different spread especially after bnt162b2, please make a correlation plot to visualize what kind of clinical parameters the people with Ab <300BAU/ml. Even if there is no correlation, seeing the data this way would be informative.

Line 259: "over 500 cells/μL (2.936 95%CI 1.961-4.394) (both

259 p<0.001) with increased antibody concentrations of 184.34% and 193.59% respectively."

Increased compared to what? people with CD4 count under 200? Please clarify.

Line 266: please call "non-response" seronegative unless there is some reason to believe this assay has an artificially high limit of detection

Line 273-6: "Being 56 to 65 years of age (p=0.025, OR 2.919) and most recent CD4+ T-cell count between 250 and 500 cells/μL or over 500 cells/μL (both p<0.001 OR 7.810 and 15.853 respectively) were significantly associated with an antibody response of more than 50 BAU/mL. Sex, viral load and nadir CD4+ T-cell count were not associated."

What about associations with being off treatment/ blips/ poor adherence or treatment type?

Line 285:

Firstly, it should be clearly explained here why nucleocapsid responses are looked at and why MOG is used as a negative control.

Secondly, the MOG responses in FigS3 are almost as big as the responses shown in S3 thus the subtraction does not make much sense. There seems to be some technical issue that has not been well described here so it's impossible to assess the value of these data. If the S3 is accurate then there is a negative control MOG response of equivalent magnitude to the Spike specific response, both before AND after vaccination. This doesn't make sense. If PLWH have an oddly higher reactivity against MOG then unstimulated control would be a better comparator. Was an unstimulated control performed? Can this data be included used to perform the subtraction instead? 

Fig 3: Cellular immune responses against SARS-CoV-2 in subgroup participants (PLWH)

There's no real indication in the data in this figure that cellular responses are worse in PLWH as claimed in conclusion

Fig 3.A

It would be clearer to have one graph for mRNA vaccines and one graph for vectored vaccines. Please indicate how many people have a neg response at T0 and T2. Consider replacing T0/T2 nomenclature by actual baseline/post 2 doses as this is more intuitive for the reader.

Fig 3.B 

Why in the left-hand panel isn't the delta response increasing with vaccination? this is whole spike peptide pool. Data so far from others shows no real alteration in WT vs VOC T cell ELISpot unless just a mutation specific pool is used. So, this seems wrong.

Line 340: "Limited data exist on SARS-CoV-2 vaccination responses in PLWH."

This is true, but there is some data in publicly available studies, and these should be cited and mentioned here to put this work into context

Line 341: "lower SARS-CoV-2 S1-specific IgG levels in PLWH compared to controls" 

Should add "as measured by diasorin assay" it's important to stress how this result was found as it is different to other studies to date.

Line 345-6: "Our primary result analysis in participants receiving mRNA vaccines contrasts with most of the small cohort studies performed in PLWH where the authors found similar responses as in controls [16-18]."

This is a major finding and so the authors should expand further on why this difference is found.

Line 350: "Most, if not all, of the previous studies had not been powered to detect a predefined size of the impact of HIV on vaccination response"

Again, here there needs to be a discussion around the fact that binding not being the be all and end all of serology and that neutralisation has not been assessed.

Line 352: "Other reasons that may explain why some of the previous studies did not find a lower response in PLWH may be the use of qualitative rather than quantitative antibody responses"

Please expand on why these results are quantitative whereas prior studies are qualitative - this doesn't seem correct (or at least intuitive) to this reviewer. 

Line 355: "However, this was observed in other vaccination studies in PLWH previously [23]"

Please highlight you mean against other pathogens

Line 357: "Our results confirm this in PLWH, in whom more immunosenescence is seen compared to HIV-negative controls [24, 25]."

Please discuss in text whether this means you are seeing these differences because cohort of PLWH contained more old people compared to other cohorts?

Line 363: "We observed a relevant IFN-γ production in the negative control stimulation in the PLWH group." 

I don't understand this sentence and I don't see the data supporting this in the figures.

Line 384: "However, at least in HIV-negative people, this can be overruled by a booster vaccination [21]."

Overruled is not the right word for this context… and can a booster fix everything? I think there needs to be more discussion about what these differences might mean for clinical vaccine efficacy and whether extra boosters or higher doses/altered regimen is needed (as per some other vaccines in PLWH)

Reviewer #3: In this work, Hensley et al. present the data from over 1154 people living with HIV/AIDS (PLWHA) and 884 controls who were vaccinated for Covid-19 to explore the anitgen-specific humoral and celluar immune responses. This is a multi-site study, involving 22 out of 24 centers providing care to PLWHA in The Netherlands.

Their findings line up with findings from other groups, suggesting that the vaccines are immunogenic and well tolerated in PLWHA. They also expand to a more detailed data on the celluar immune response, which is a strong point, despite the number of analyzed subjects.

The results adds, in a compreehensive observation, the need to vaccinate PLWHA for Covid-19. Although no efficacy data is presented and one still lack the marker of protection for the disease, this is important information to be generated and communicated.

Minor comment:

1. Authors should acknowledge findings with other vaccine platforms, including inactivated virus

[LINK]

---

## [Decision Letter · Decision Letter 2]

26 Jul 2022

Dear Dr. Hensley,

Thank you very much for submitting your manuscript "Immunogenicity and reactogenicity of SARS-CoV-2 vaccines in people living with HIV in the Netherlands: a nationwide prospective cohort study" (PMEDICINE-D-22-01021R2) for consideration at PLOS Medicine. 

[LINK]

In light of these reviews, I am afraid that we will not be able to accept the manuscript for publication in the journal in its current form, but we would like to consider a revised version that addresses the reviewers' and editors' comments. Obviously we cannot make any decision about publication until we have seen the revised manuscript and your response, and we plan to seek re-review by one or more of the reviewers. 

We expect to receive your revised manuscript by Aug 16 2022 11:59PM. Please email us (plosmedicine@plos.org) if you have any questions or concerns.

We look forward to receiving your revised manuscript. 

Sincerely,

Beryne Odeny, 

PLOS Medicine

plosmedicine.org

Comments from the Academic editor:

Just noting that in the legend of Fig 2 the authors need to state the p values (and test used) for comparisons between HIV- and PLWH, and how many subjects in each group (they have included this information in the Figure 3 legend). And also state what the numbers above the dot-plots represent.

Comments from the reviewers:

Reviewer #1: Many thanks to the authors for satisfactorily considering and responding to each comment in turn, amending the manuscript where necessary.

Reviewer #2: The major original critique was that the T cell component was confusing. This has not been well-addressed. It is still unclear if the spike SFU numbers shown in FigS4 have already had the background subtracted or not, I would strongly suggest DMSO subtraction is more comparable to other studies than MOG subtraction. The data in FigS3 look as expected but it doesn't make sense how they are derived from the data in FigS4 as the Spike responses are equal in magnitude to the DMSO/MOG/NC in Fig4S? And the DMSO responses are abnormally large. This is very different to other studies where spike-specific responses are much larger than DMSO - this is how we know the assay has actually worked when running them. Based on the data presented I cannot support their publication as I am concerned there is an artefact, in my view these data should be removed as the serology is valid independently.

The issue of referring to a non-response rather than seronegative has not been resolved, I think this would be clearer for the wider community doing a variety of assays not just this specific one. Also, they refer to clinical expertise in drawing the line at 50 BAU/ml where the manufacturer says 33.8 BAU/ml. I understand views on what is a "useful" response may be taken in a clinical setting but it doesn't make clear sense to me in a research paper to disregard manufacturers guidelines on cut-offs based on sensitivity and specificity data using known positive and negative cases that enabled clinical use of the assay in the first place and call this "non-response". Moreover, the term "adequate response" for >300 BAU/ml is misleading as it implies some knowledge of how this assay relates to protection which has not been explored here. I would strongly suggest using the term "normal" rather than "adequate".

The new Author summary also has a major issue in the following claim:

" Because the height of the antibody response correlates with protection against symptomatic infection"

This paper does not look at protection in anyway, so it is not a good idea to make this claim here. If there are papers to support that binding responses (especially this particular assay and the cut offs used by these investigators) correlate with protection from symptomatic infection, then these should be discussed in the discussion and not presented as a key finding /outcome of this piece of work in the author summary. The discussion presents a set of studies linking various parameters to one another but not the direct link of binding to protection, and importantly not the magnitude of binding response in this assay to protection so do not support this statement, and regardless I don't think it belongs in the Author Summary as it is not one of their findings.

[LINK]

---

## [Decision Letter · Decision Letter 3]

12 Sep 2022

Dear Dr. Hensley,

Thank you very much for re-submitting your manuscript "Immunogenicity and reactogenicity of SARS-CoV-2 vaccines in people living with HIV in the Netherlands: a nationwide prospective cohort study" (PMEDICINE-D-22-01021R3) for review by PLOS Medicine.

I have discussed the paper with my colleagues and the academic editor and it was also seen again by one reviewer. I am pleased to say that provided the remaining editorial and production issues are dealt with we are planning to accept the paper for publication in the journal.

[LINK]

We look forward to receiving the revised manuscript by Sep 19 2022 11:59PM.   

Sincerely,

Beryne Odeny, 

PLOS Medicine

plosmedicine.org

Requests from Editors:

1) The Data Availability Statement (DAS) requires revision for each data source used in your study. If the data are not freely available, please include an appropriate contact (web or email address) for inquiries (this cannot be a study author).

2) In the supplementary files, please indicate p <0.001 instead of p = 0.000

Comments from Reviewers:

Reviewer #2: Acceptable clarifications have now been made to the T cell part of the story.

[LINK]

---

## [Editor Report · Decision Letter 4]

20 Sep 2022

Dear Dr Hensley, 

On behalf of my colleagues and the Academic Editor, Dr. James G. Beeson, I am pleased to inform you that we have agreed to publish your manuscript "Immunogenicity and reactogenicity of SARS-CoV-2 vaccines in people living with HIV in the Netherlands: a nationwide prospective cohort study" (PMEDICINE-D-22-01021R4) in PLOS Medicine.

PRESS

Sincerely, 

Beryne Odeny 

PLOS Medicine